# Fair Active Learning in Low-Data Regimes

**Romain Camilleri, Andrew Wagenmaker, Jamie Morgenstern, Lalit Jain, Kevin Jamieson**

University of Washington, Seattle, WA
`{camilr,ajwagen,jamiemmt,jamieson}@cs.washington.edu,lalitj@uw.edu`

## Abstract

In critical machine learning applications, ensuring fairness is essential to avoid perpetuating social inequities. In this work, we address the challenges of reducing bias and improving accuracy in data-scarce environments, where the cost of collecting labeled data prohibits the use of large, labeled datasets. In such settings, active learning promises to maximize marginal accuracy gains of small amounts of labeled data. However, existing applications of active learning for fairness fail to deliver on this, typically requiring large labeled datasets, or failing to ensure the desired fairness tolerance is met on the population distribution.

To address such limitations, we introduce an innovative active learning framework that combines an exploration procedure inspired by posterior sampling with a fair classification subroutine. We demonstrate that this framework performs effectively in very data-scarce regimes, maximizing accuracy while satisfying fairness constraints with high probability. We evaluate our proposed approach using well-established real-world benchmark datasets and compare it against state-of-the-art methods, demonstrating its effectiveness in producing fair models, and improvement over existing methods.

## 1 INTRODUCTION

As machine learning models proliferate and are used in an ever-increasing number of applications with societal ramifications, it has become increasingly important to have robust methods for developing models that do not perpetuate existing social inequities. Over the last few years, a plethora of works in fair classification have provided a principled toolkit to develop classifiers and quantify their performance under various fairness metrics. These metrics,

including equal opportunity and equalized odds, give a natural way to ensure that favorable outcomes such as model performance or predicted positive rates are equalized across different groups for a given protected feature. More precisely, given a distribution $\nu$ on $\mathcal{X} \times \mathcal{A} \times \mathcal{Y}$ (where $\mathcal{X}$ is the feature space, $\mathcal{A}$ the protected attribute space and $\mathcal{Y}$ the label space), a hypothesis class $\mathcal{H}$, a fairness metric $m_{\text{fair}}$, a measure of its violation $L_\nu^{m_{\text{fair}}}(h)$, and a fairness violation tolerance $\alpha$; the goal in fair classification is to return $\arg\min_{h \in \mathcal{H}} \mathbb{E}_{(x,a,y)\sim\nu}[h(x) \neq y]$ subject to $L_\nu^{m_{\text{fair}}}(h) \leq \alpha$.

In practice, as $\nu$ is unknown, solving an empirical analog of this constrained classification problem on a training set is a natural approach to learning classifiers that generalize well to a test set, while maintaining fairness guarantees. Indeed, the focus of much of the fairness literature has been to develop optimization methods to solve such a problem (Agarwal et al., 2018; Cotter et al., 2018; Donini et al., 2018). While this is a reasonable approach when a large amount of labeled training data is available, in many applications such large amounts of data are not available, and it can be prohibitively expensive to collect more. In such settings existing approaches may not be able to guarantee accurate classifiers, or may return classifiers that are in fact unfair on the population distribution.

A promising approach to handle such low-data regimes and maximize the effectiveness of small amounts of labeled data is *active learning*. Active learning methods aim to minimize the amount of labeled training data needed by only requesting labels for the most *informative* examples, thereby significantly reducing the label complexity while ensuring similar accuracy of the learned classifier. While active learning methods have been applied to fair classification before, existing works either require large labeled datasets for pre-training, thereby eliminating the primary benefit of active learning, or are unable to satisfy the goal fairness constraint.

In this work we aim to overcome these challenges and develop methods for fair active learning which do not require

large pretraining datasets—truly operating in the low-data regime—and ensure fairness constraints are met. Our contributions are as follows:

1. We propose a novel approach to fair active learning, FARE, which chooses which points to label by combining a posterior sampling-inspired randomized exploration procedure that aims to improve classifier accuracy, with a group-dependent sampling procedure to ensure fairness is met. Notably, our approach does not require a large pretraining dataset, and is able to produce accurate and fair classifiers in the very low data regime.

2. We evaluate our proposed method on a variety of standard benchmark datasets from the fairness community, and demonstrate that it yields large label complexity gains over passive approaches while ensuring fairness constraints are met, and also significantly outperforms the existing state-of-the-art approaches for fair active learning.

To the best of our knowledge, our proposed approach is the first active learning procedure able to ensure fairness constraints are reliably met without requiring large amounts of labeled data.

## 2 RELATED WORK

**Fairness.** Algorithmic fairness has garnered significant interest in recent years (see Barocas et al. (2017); Hort et al. (2022) for recent surveys). Approaches to mitigate fairness disparities can be grouped into three lines of work: pre-processing, in-processing, and post-processing. Pre-processing aims to remove disparate impact by modifying the training data(Kamiran and Calders, 2012), while post-processing modifies already learned classifiers to improve fairness (Hardt et al., 2016). Of particular interest to our work is in-processing for bias mitigation, where the focus is on modifying the learning process to build fair classifiers (Zhang et al., 2018). Most relevant to us within in-processing bias mitigation techniques are works that have approached fairness mitigations in classification as a constrained optimization problem (Agarwal et al., 2018; Donini et al., 2018). Our fairness metrics of interest—equal opportunity and equalized odds—were introduced as operationalizations of fairness concurrently by Hardt et al. (2016); Kleinberg et al. (2016); see also Kearns et al. (2018).

**Active learning.** The expense associated with labeling data has emerged as a significant obstacle in the practical implementation of machine learning methods. Motivated by this, there has been growing attention towards the concept of *active* classification, which involves presenting the learner with a set of unlabeled examples, and tasking them with producing a precise hypothesis after querying as few labels as possible (Settles, 2011). Active learning has been studied extensively over the past five decades (see the survey Hanneke (2014)). Most active learning approaches select

samples to label based on some notion of uncertainty (e.g., entropy of predictions, margin, disagreement (Beygelzimer et al., 2009; Cohn et al., 1994)). Recent breakthroughs have connected best-arm identification for linear bandits with classification, opening up new possibilities for active learning via *experiment design* (Camilleri et al., 2022, 2021; Katz-Samuels et al., 2021).

**Fair active learning.** The problem of fair active classification has been previously considered by recent efforts to reach a classifiers with good "fairness-error" trade-off given a label budget, including Anahideh et al. (2021); Fajri et al. (2022); Sharaf et al. (2022). As we will see experimentally, these works suffer from a variety of shortcomings: for example, poor generalization of their fairness violation, minimal accuracy gains over baseline methods, or limited ability to handle standard group fairness metrics. Furthermore, their objective is somewhat different than ours. While we aim to return a classifier with fairness violation below a desired tolerance (motivated by situations where it is critical to ensure our classifier satisfies a given fairness constraint), these works instead aim to quantify the general tradeoff between fairness and accuracy, without ensuring the returned classifier is below any tolerance. Last, these works assume the existence of large, pre-existing, labeled datasets: namely for their experiments on the `Adult income` dataset Anahideh et al. (2021); Fajri et al. (2022); Sharaf et al. (2022) assume respectively that 2000, 15000, 3000 labels are accessible. We will see that the gains from our active learning algorithms are instead visible after collecting 100 labels. Other works, such as Cao and Lan (2022b) focuses on fair active learning for decoupled models and Cao and Lan (2022a); Shen et al. (2022), have focused on the analogous problem of finding classifiers that meet *metric*-fair constraints, while Abernethy et al. (2020); Branchaud-Charron et al. (2021); Cai et al. (2022); Shekhar et al. (2021) have focused on data collection for *min-max* fairness. The nature of min-max fairness does not explicitly constrain the differences in quantities between groups, instead improving the quantity for the worst-off group as much as possible. These, alongside the metric fairness constraints, are significantly different than the group fairness metrics we consider, and as such motivate an entirely different set of methods.

Another related line of work is that of bandits with constraints (Camilleri et al., 2022; Kazerouni et al., 2017; Pacchiano et al., 2021; Sui et al., 2015; Wang et al., 2022). As noted, classification can be modeled as a bandit problem and in some cases bandit algorithms can be applied to active learning for classification. Furthermore, imposing unknown constraints in bandit problems is similar to imposing fairness constraints in classification. To the best of our knowledge, however, existing work on constrained bandits does not consider constraints expressive enough to encode standard fairness metrics such as equalized odds and equal opportunity.

# 3 PRELIMINARIES

In this work, we focus on a binary classification scenario where each data point consists of three elements $(x, a, y)$. Here, $x \in \mathcal{X} \subset \mathbb{R}^d$ represents a $d$-dimensional feature vector, $a \in \{0, 1\}$ indicates a binary protected attribute which partitions our data into two *groups*, and $y \in \{0, 1\}$ denotes a label. In the general classification paradigm, we assume that the training set $\mathcal{D} = \{(x_1, a_1, y_1), \ldots, (x_n, a_n, y_n)\} \sim \nu \in \triangle_{\mathcal{X} \times \{0,1\} \times \{0,1\}}$ is a set of $n$ examples sampled from a target distribution $\nu$. The objective is to learn from the training set $\mathcal{D}$ a classifier $h : \mathcal{X} \mapsto \{0, 1\}$ among a hypothesis set $\mathcal{H}$ (e.g. linear classifiers or random forests) which has the lowest risk $R_\nu(h)$ possible on the target distribution. Here the risk is defined for any distribution $\nu \in \triangle_{\mathcal{X} \times \{0,1\} \times \{0,1\}}$ as $R_\nu(h) := \mathbb{E}_{(x,a,y)\sim\nu}[\mathbb{1}\{h(x) \neq y\}]$.

## 3.1 DEFINITIONS OF FAIRNESS

In this work we consider in particular two well-known definitions of fairness: Equal Opportunity—also called True Positive Rate Parity (TPRP)—and Equalized Odds (EO), though our method extends to other notions of fairness as well. We formally define these here.

**Definition 1** (Fairness Definitions (EO, TPRP)). *Given a tolerance $\alpha \in [0, 1]$ and target distribution $\nu$, a classifier $h \in \mathcal{H}$ satisfies True Positive Rate Parity up to $\alpha$ on $\nu$ if*

$$|P_{(x,a,y)\sim\nu}(h(x) = 1|a = 0, y = 1)$$
$$- P_{(x,a,y)\sim\nu}(h(x) = 1|a = 1, y = 1)| \leq \alpha. \quad (1)$$

*A classifier satisfies Equalized Odds up to $\alpha$ on a distribution $\nu$ if, in addition to satisfying* (1) *it also satisfies*

$$|P_{(x,a,y)\sim\nu}(h(x) = 1|a = 0, y = 0)$$
$$- P_{(x,a,y)\sim\nu}(h(x) = 1|a = 1, y = 0)| \leq \alpha. \quad (2)$$

If $\alpha = 0$, EO states that the prediction $h(x)$ is conditionally independent of the protected attribute $a$ given the label $y$. With these definitions of fairness in mind, we also define the fairness violation of a given classifier as the left-hand sides of equations (1) and (2).

**Definition 2** (Fairness violation). *We define the EO (resp. TPRP) violation of classifier h on distribution $\nu$ as*

$$L_\nu^{\text{EO}}(h) := \max_{z \in \{0,1\}} |P_{(x,a,y)\sim\nu}(h(x) = 1|a = 0, y = z)$$
$$- P_{(x,a,y)\sim\nu}(h(x) = 1|a = 1, y = z)|,$$
$$L_\nu^{\text{TP}}(h) := |P_{(x,a,y)\sim\nu}(h(x) = 1|a = 0, y = 1)$$
$$- P_{(x,a,y)\sim\nu}(h(x) = 1|a = 1, y = 1)|.$$

Given some threshold $\alpha$, a *fair classifier* is a classifier with fairness violation below $\alpha$.

## 3.2 PROBLEM STATEMENT

Classical machine learning typically deals with the setting where the learner has access to a fixed, labeled dataset, $\mathcal{D}_{\text{tr}}$, and must learn as accurate a classifier as possible from this data. In this work, we are interested in the *active* setting where the goal of the learner is to train on as few labeled data points as possible to obtain a desired accuracy. In particular, in the pool-based active learning setting, the task of fair active classification is the following sequential problem. First, the learner is given an unlabeled training pool of data $\mathcal{D}_{\text{tr}}^{\backslash y} \subseteq \mathcal{X} \times \mathcal{A}$ and some fairness metric $m_{\text{fair}} \in \{EO, TP\}$ with target fairness violation $\alpha$. At each time $t = 1, 2, \ldots, T$ the agent then chooses any unlabeled point from the pool $(x_t, a_t) \in \mathcal{D}_{\text{tr}}^{\backslash y}$ and requests its label $y_t \in \{0, 1\}$. After requesting $T$ labels, the agent outputs a classifier $h \in \mathcal{H}$. Its performance is evaluated via the two following metrics: error loss $R_\nu(h)$ and fairness violation $L_\nu^{m_{\text{fair}}}(h)$, for $\nu$ the population distribution. Note that we assume that the learner may see the true protected attribute before querying the label for a point—see Awasthi et al. (2020) for a discussion of the case when the protected attribute is noisy.

# 4 FAIR ACTIVE LEARNING

In this section, we present our approach to fair active classification, FARE.

## 4.1 FAIR LEARNING WITH FIXED DATASETS

Before considering the active setting, we first consider the question of finding a fair classifier on a fixed dataset. As the general classification paradigm (i.e. classification without fairness constraints) is known to potentially cause disparities when applied to sensitive tasks (Barocas and Selbst, 2016), significant effort has been invested to develop effective algorithms that balance the goal of classification (learn the most accurate classifier) with fairness (learn a classifier with low fairness violation) on static datasets. Given a target distribution $\nu$, a fairness metric denoted $m_{\text{fair}} \in \{EO, TP\}$ and a fairness violation tolerance $\alpha \in [0, 1]$, this fair classification problem can be stated as the following:

$$\underset{h \in \mathcal{H}}{\text{minimize}} \quad R_\nu(h) \quad \text{subject to} \quad L_\nu^{m_{\text{fair}}}(h) \leq \alpha. \quad (3)$$

In practice, one cannot solve (3) directly, as the population, $\nu$, which $R_\nu(h)$ and $L_\nu^{m_{\text{fair}}}(h)$ depend on, is unknown. Instead, we consider empirical estimates of the risk and fairness constraint. As is standard throughout machine learning, we rely on the plug-in estimate of the empirical risk, $\hat{R}_\mathcal{D}(h) := \frac{1}{n} \sum_{i=1}^n \mathbb{1}\{h(x_i) \neq y_i\}$. Similarly, throughout the fairness literature, a plug-in estimator is typically also used to estimate the fairness violation (Agarwal et al., 2018; Cotter et al., 2018; Donini et al., 2018). As an example, consider the case of estimating TPRP. Let $\mathcal{D} = \{(x_1, a_1, y_1), \ldots, (x_n, a_n, y_n)\}$ denote a set of data and recall that the True Positive Rate (TPR) of each group $z \in \{0, 1\}$ can be written as

$$P_{(x,a,y)\sim\nu}(h(x) = 1|a = z, y = 1)$$
$$= \frac{\mathbb{E}_{(x,a,y)\sim\nu}[\mathbb{1}\{h(x) = 1, y = 1, a = z\}]}{\mathbb{E}_{(x,a,y)\sim\nu}[\mathbb{1}\{y = 1, a = z\}]}. \quad (4)$$

A natural approach to empirically estimate the TPRP is then to simply replace the population quantities with the empirical quantities in (4) to estimate the TPR for each group, and then compute the absolute value of the difference of these TPRs. This yields the following empirical estimate of the TPRP violation of a classifier $h$ on the data $\mathcal{D}$:

$$\widehat{L}_{\mathcal{D}}^{\text{TP}}(h) := \left| \sum_{i=1}^{n} \frac{\mathbb{1}\{h(x_i)=1, y_i=1, a_i=1\}}{\sum_{i=1}^{n} \mathbb{1}\{y_i=1, a_i=1\}} \right.$$
$$\left. - \sum_{i=1}^{n} \frac{\mathbb{1}\{h(x_i)=1, y_i=1, a_i=0\}}{\sum_{i=1}^{n} \mathbb{1}\{y_i=1, a_i=0\}} \right|. \quad (5)$$

We can estimate the *false-positive rate parity* (FPRP), $\widehat{L}_{\mathcal{D}}^{\text{FP}}(h)$, analogously to (5) but with $y_i = 1$ replaced by $y_i = 0$, and estimate the EO violation as the maximum of the empirical estimate of the TPRP violation and the empirical estimate of the FPRP violation, $\widehat{L}_{\mathcal{D}}^{\text{EO}}(h) = \max\{\widehat{L}_{\mathcal{D}}^{\text{TP}}(h), \widehat{L}_{\mathcal{D}}^{\text{FP}}(h)\}$.

**Empirical fair classification.** Equipped with these empirical estimates, we return to the fair classification problem, (3). Given a training set $\mathcal{D} = \{(x_1, a_1, y_1), \dots, (x_n, a_n, y_n)\} \sim \nu$ sampled from a distribution $\nu \in \triangle_{\mathcal{X} \times \{0,1\} \times \{0,1\}}$, a fairness metric denoted $m_{\text{fair}} \in \{EO, TP\}$ and fairness tolerance $\alpha \in [0, 1]$, one can use the empirical estimates of the risk and the fairness violation to approximate (3) with the following *empirical fair classification optimization problem*:

$$\underset{h \in \mathcal{H}}{\text{minimize}} \quad \widehat{R}_{\mathcal{D}}(h) \quad \text{subject to} \quad \widehat{L}_{\mathcal{D}}^{m_{\text{fair}}}(h) \le \alpha. \quad (6)$$

Note that solving such a problem is a common approach to fair classification, and can be solved efficiently (Agarwal et al., 2018; Donini et al., 2018). This optimization problem will form the starting-point of our proposed approach, and our algorithms will assume access to a solver for it, which we call the empirical fair oracle—EFO. In our experiments we take an approach analogous to Agarwal et al. (2018) to solve (6).

## 4.2 ESTIMATION ERROR AND SAMPLING BIAS

In this section we address two additional issues that arise in ensuring our returned classifier is fair. First, estimation error in the fairness constraint, and second, bias introduced by sampling data points in a non-uniform fashion.

**Conservative fairness estimates.** Since $\widehat{L}_{\mathcal{D}}^{m_{\text{fair}}}(h)$ is only an empirical estimate of $L_{\nu}^{m_{\text{fair}}}(h)$, ensuring that $\widehat{L}_{\mathcal{D}}^{m_{\text{fair}}}(h) \le \alpha$ does not guarantee that $L_{\nu}^{m_{\text{fair}}}(h) \le \alpha$, our end goal. The following result gives a precise quantification of the deviation between $\widehat{L}_{\mathcal{D}}^{m_{\text{fair}}}(h)$ and $L_{\nu}^{m_{\text{fair}}}(h)$ in the case where $m_{\text{fair}} = EO$.

**Proposition 4.1.** *Let the train set be $\mathcal{D} = \{(x_1, a_1, y_1), \dots, (x_n, a_n, y_n)\} \sim \nu$. Then it holds*

*with probability $1 - \delta$ that, with $c_\delta := 8\sqrt{2\log(2/\delta)}$:*

$$|L_{\nu}^{\text{EO}}(h) - \widehat{L}_{\mathcal{D}}^{\text{EO}}(h)|$$
$$\le \frac{c_\delta}{\sqrt{n}} \cdot \max_{0 \le j,k \le 1} \frac{1}{\frac{1}{n}\sum_{i=1}^{n} \mathbb{1}\{y_i=k, a_i=j\}} + \mathcal{O}\left(\frac{1}{n}\right).$$

Analogous results hold for TPRP. This bound inspires two important aspects of our approach. First, to ensure fairness is met, it suggests setting the tolerance in (6) to a conservative value less than $\alpha$, in particular subtracting a $\mathcal{O}(\frac{1}{\sqrt{n}})$ term off of $\alpha$. Adjusting $\alpha$ by this margin has been demonstrated in the past to produce fair classifiers (Thomas et al., 2019; Woodworth et al., 2017), and we show in Figure 10 that it is also critical in our active setting. Second, Proposition 4.1 suggests that in order to estimate the fairness, we need to collect samples for *each protected attribute*, since our estimation error scales inversely with the minimum number of samples collected for either protected attribute. This observation is critical in motivating our active sampling procedure, as we outline in the following section.

**Sampling bias correction.** In the active learning paradigm, at every step the learner samples a data point $(x_t, a_t) \in \mathcal{D}_{\text{tr}}^{\setminus y}$ from some (chosen) distribution, $\nu_t^{\text{tr}} \in \triangle_{\mathcal{D}_{\text{tr}}^{\setminus y}}$, $(x_t, a_t) \sim \nu_t^{\text{tr}}$. For example, the learner may place higher weight on points that are *informative*, increasing the number of samples from around the decision boundary. While this will ultimately improve the learner's ability to classify, the distribution of the sampled dataset no longer matches that of the original training dataset. This will result in the plug-in estimator for the fairness constraint, for example (5), to be biased. We correct for this mismatch using importance weights. For the risk, we recall the definition of the well-known IPS estimator (empirical risk re-weighted with importance weights): $\widehat{R}_{\mathcal{D}, \nu^{\text{tr}}, \nu}(h) := \frac{1}{n}\sum_{i=1}^{n} \frac{\nu_i}{\nu_i^{\text{tr}}} \mathbb{1}\{h(x_i) \ne y_i\}$, for $(x_i, a_i) \sim \nu^{\text{tr}}$ and $y_i$ and associated label, and $\nu_i$ the population weight of point $i$[1] and $\nu_i^{\text{tr}}$ the probability $\nu^{\text{tr}}$ samples point $i$. It is straightforward to see that this is an unbiased estimator of the true risk. We define the estimator for EO with importance weights next.

**Definition 3** (Empirical EO violation with importance weights). *Consider a dataset drawn i.i.d from $\nu^{\text{tr}}$, $\mathcal{D} := \{(x_1, a_1, y_1), \dots, (x_n, a_n, y_n)\} \sim \nu^{\text{tr}}$. The empirical estimate of the EO violation of a classifier $h$ on the target distribution $\nu$ can be evaluated as*

$$\widehat{L}_{\mathcal{D}, \nu^{\text{tr}}, \nu}^{\text{EO}}(h) := \max_{z \in \{0,1\}} \left| \frac{\sum_{i=1}^{n} \frac{\nu_i}{\nu_i^{\text{tr}}} \mathbb{1}\{h(x_i)=1, y_i=z, a_i=1\}}{\sum_{i=1}^{n} \frac{\nu_i}{\nu_i^{\text{tr}}} \mathbb{1}\{y_i=z, a_i=1\}} \right.$$
$$\left. - \frac{\sum_{i=1}^{n} \frac{\nu_i}{\nu_i^{\text{tr}}} \mathbb{1}\{h(x_i)=1, y_i=z, a_i=0\}}{\sum_{i=1}^{n} \frac{\nu_i}{\nu_i^{\text{tr}}} \mathbb{1}\{y_i=z, a_i=0\}} \right|.$$

---

[1]In general this is unknown but, assuming $\mathcal{D}_{\text{tr}}^{\setminus y} \sim \nu$, it suffices to simply set $\nu_i = 1/|\mathcal{D}_{\text{tr}}^{\setminus y}|$

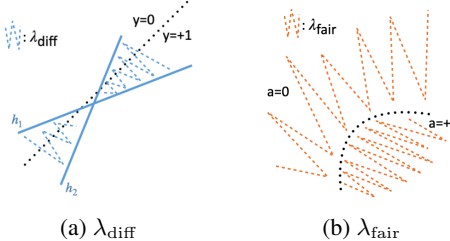

(a) $\lambda_{\text{diff}}$         (b) $\lambda_{\text{fair}}$

Figure 1: Sampling distributions of FARE when $k = 2$. The oscillating dotted lines are used to represent the support of the sampling distributions (areas where the sampling distribution is non-zero). $\lambda_{\text{diff}}$ places mass on disagreement region of learned classifiers in order to collect points increasing accuracy. $\lambda_{\text{fair}}$ places equal amounts of mass on each group in order to learn fairness value.

We define the importance-weighted TPRP violation analogously, but only for $z = 1$. While this estimate is not truly unbiased, both the numerator and denominators are unbiased, leading to accurate estimates of the fairness. In the following, when applying our fairness oracle EFO in the active setting, we assume it is applied on the importance-weighted fairness and loss estimates.

### 4.3 FAIR ACTIVE LEARNING

We now provide our algorithm for fair active classification, Algorithm 1. Algorithm 1 proceeds in rounds. In each round, we choose data points to label by sampling from two distributions: $\lambda_{\text{diff}}$, which focuses on improving the *accuracy*, and $\lambda_{\text{fair}}$, which focuses on improving the *fairness estimates*. We describe our choice of each of these distributions below.

**Improving accuracy via randomized exploration.** In each round of Algorithm 1, to determine which points are most likely to improve accuracy, we perform *randomized exploration* by training a set of $k$ fair classifiers $\widehat{h}_i, i \in [k]$, on perturbations of the training data already collected. In particular, to generate these perturbations, while training each classifier $\widehat{h}_i$ we flip the label of each data point with probability $\sigma$. Given these classifiers, we compute $\lambda_{\text{diff}}$, which aims to sample unlabeled training points that effectively distinguish between the $k$ classifiers.

As described in a variety of works (Camilleri et al., 2022; Kveton et al., 2019; Osband et al., 2018, 2019, 2016; Russo, 2019), randomized exploration emulates sampling from a posterior distribution over the optimal classifier. The sampling distribution $\lambda_{\text{diff}}$ is such that the weights will be large for the points $x$ about which the $k$ classifiers disagree most. Indeed, taking $k = 2$ for illustration, we have $\lambda_{\text{diff}} = \arg\min_{\lambda \in \triangle_{\mathcal{X}}} \sum_{x \in \mathcal{X}} \frac{\mathbb{1}\{h_1(x) \neq h_2(x)\}}{\lambda_x}$. If $h_1(x) = h_2(x)$ then $\frac{\mathbb{1}\{h_1(x) \neq h_2(x)\}}{\lambda_x} = 0$ for any $\lambda_x > 0$. In order to minimize $\sum_{x \in \mathcal{X}} \frac{\mathbb{1}\{h_1(x) \neq h_2(x)\}}{\lambda_x}$, one can set $\lambda_x$ to be very small at regions of $\mathcal{X}$ where $h_1 = h_2$ and very large at re-

---

**Algorithm 1** FARE (Fair Active Randomized Exploration)

**Require:** Batch size $n$, number of rounds $L$, classifiers per round $k$, perturbation rate $\sigma$, fairness metric $m_{\text{fair}}$, fairness tolerance $\alpha$, unlabeled data $\mathcal{D}_{\text{tr}}^{\setminus y}$

1: Sample $(x_1^{(0)}, a_1^{(0)}), \ldots, (x_n^{(0)}, a_n^{(0)}) \sim \text{Unif}(\mathcal{D}_{\text{tr}}^{\setminus y})$, request labels for sampled points
2: $\mathcal{D}_0 \leftarrow \{(x_i^{(0)}, a_i^{(0)}, y_i^{(0)})\}_{i=1}^n$
3: $\mathcal{D}_{\text{tr}}^{\setminus y} \leftarrow \mathcal{D}_{\text{tr}}^{\setminus y} \setminus \{(x_i^{(0)}, a_i^{(0)})\}_{i=1}^n$
4: **for** $\ell = 1, \ldots, L-1$ **do**
   // Compute $\lambda_{\text{diff}}$
5:     **for** $i = 1, \ldots, k$ **do**
6:        $h_i = \text{EFO}(\widetilde{\mathcal{D}}_{\ell-1}, \alpha - \frac{1}{\sqrt{n \cdot \ell}})$ where $\widetilde{\mathcal{D}}_{\ell-1}$ generated by flipping each label of $\mathcal{D}_{\ell-1}$ w.p. $\sigma$
7:     **end for**
8:     Compute $\lambda_{\text{diff}}$ allocation:

$$\lambda_{\text{diff}} \leftarrow \arg\min_{\lambda \in \triangle_{\mathcal{D}_{\text{tr}}^{\setminus y}}} \max_{1 \leq i \neq j \leq k} \sum_{(x,a) \in \mathcal{D}_{\text{tr}}^{\setminus y}} \frac{\mathbb{1}\{h_i(x) \neq h_j(x)\}}{\lambda_x}$$

   // Compute $\lambda_{\text{fair}}$
9:     $\lambda_{\text{fair}} \leftarrow \frac{1}{2}\text{Unif}(\{(x,a) \in \mathcal{D}_{\text{tr}}^{\setminus y} : a = 0\})$
              $+ \frac{1}{2}\text{Unif}(\{(x,a) \in \mathcal{D}_{\text{tr}}^{\setminus y} : a = 1\})$
   // Sample points and update classifier
10:    Sample $(x_i^{(\ell)}, a_i^{(\ell)}) \sim \frac{1}{2}\lambda_{\text{diff}} + \frac{1}{2}\lambda_{\text{fair}}, i = 1, \ldots, n$
11:    Observe corresponding labels $y_1^{(\ell)}, \ldots, y_n^{(\ell)}$
12:    $\mathcal{D}^\ell \leftarrow \mathcal{D}^{\ell-1} \cup \{(x_i^{(\ell)}, a_i^{(\ell)}, y_i^{(\ell)})\}_{i=1}^n$
13:    $\mathcal{D}_{\text{tr}}^{\setminus y} \leftarrow \mathcal{D}_{\text{tr}}^{\setminus y} \setminus \{(x_i^{(\ell)}, a_i^{(\ell)})\}_{i=1}^n$
14: **end for**
15: **Return** $\widehat{h} = \text{EFO}(\mathcal{D}^L, \alpha - \frac{1}{\sqrt{n \cdot L}})$

---

gions of $\mathcal{X}$ where $h_1 \neq h_2$. See Figure 1a for an illustration of this. Given this, if we can ensure $\widehat{h}_i, i \in [k]$ disagree on points close to the true decision boundary, then our sampling procedure will ensure that we sample such points, which will enable us to effectively learn an accurate classifier. With this in mind, we hope to create $k$ classifiers that have a decision boundary close to the true decision boundary, yet this is precisely what will be created by posterior sampling, which our procedure mimics. As we will see in the experiments, this sampling strategy effectively collects labels that are informative, increasing accuracy of the learned classifier.

**Improving fairness via attribute-dependent exploration.** In addition to learning the decision boundary to obtain a classifier with high accuracy, we must also learn the value of the fairness constraint to ensure our final classifier is fair. While $\lambda_{\text{diff}}$ ensures that we sample points close to the decision boundary, it makes no guarantee that we sample points which allow us to accurately estimate our fairness constraint—our choice of $\lambda_{\text{fair}}$ ensures that we do sample enough to accurately estimate the fairness.

As shown in Proposition 4.1, if we wish to estimate the

| Dataset | Protected Attribute | Dataset Size |
|---|---|---|
| `Drug Consumption` (Fehrman et al., 2017) | Gender | 1885 |
| `Bank` (Moro et al., 2014) | Education Level | 11,162 |
| `German Credit` (Hofmann, 1994) | Gender | 1,000 |
| `Adult Income` (Lichman, 2013) | Gender | 48,842 |
| `Compas` (Lichman, 2013) | Gender | 5,278 |
| `Community and Crime` (Redmond and Baveja, 2002) | Race | 1,902 |

Table 1: Benchmark datasets

fairness value of a given classifier, we must ensure that we have collected sufficiently many data points from each group $j \in \{0, 1\}$. $\lambda_{\text{diff}}$ is not guaranteed to sample such points—for example, if we have severe group imbalance, the overall accuracy may be maximized by ignoring the group with many fewer samples, in which case $\lambda_{\text{diff}}$ will focus on only sampling the larger group. To address this, we choose $\lambda_{\text{fair}}$ to sample an equal number of samples from each group, which will ensure that our fairness estimate will converge to the population fairness, as guaranteed by Proposition 4.1. See Figure 1b for an illustration of this. As we demonstrate in Section 5.3, this sampling is absolutely critical if our goal is to learn a fair classifier—without this attribute-dependent sampling, naive active learning methods fail to produce fair classifiers.

# 5 EXPERIMENTS

Finally, we demonstrate the effectiveness of FARE experimentally on standard fairness datasets.

**Implementation details.** For all experiments, we use logistic regression classifiers without regularization and partition the dataset into a $75\%/25\%$ train/test split. We ran a grid-search over the hyperparameters of FARE to set $\sigma = 0.1$ and $k = 10$. We set the fairness tolerance to $\alpha - 1/\sqrt{n}$ to account for estimation error in the fairness constraint. All experiments were run on a Intel Xeon 6226R CPU with 64 cores.

**Datasets.** In our experiments, we consider six datasets commonly used in the fairness literature, listed in Table 1. To ensure consistency, we standardized the data to have a mean of zero and a variance of one.

## 5.1 BASELINES METHODS

In order to benchmark FARE, we conduct experiments comparing it against state-of-the-art algorithms (Anahideh et al., 2021; Fajri et al., 2022; Sharaf et al., 2022) for fair active learning, and a passive baseline.

1. PANDA (Sharaf et al., 2022): PANDA aims to learn a data selection policy via meta-learning. This algorithm formulates the problem as a bi-level optimization task, where the inner level involves training a classifier with a subset of labeled data, while the outer level focuses on updating the selection policy to strike a balance between fairness and accuracy in the classifier's performance.

2. FAL (Anahideh et al., 2021): FAL uses a sampling rule that blends between two selection criteria: one based on uncertainty and another based on assessing fairness, which estimates the potential disparity impact when labeling a specific data point (by calculating the expected disparity across all potential labels). FAL chooses which data points to label in order to strike a balance between model accuracy and equity.

3. FALCUR (Fajri et al., 2022): FALCUR incorporates an acquisition function that assesses the representative score of each sample under consideration. This score is calculated by taking into account two key factors: uncertainty and similarity. By carefully balancing these elements, FALCUR selects samples that contribute to accuracy improvement and ensure that fairness is maintained.

4. Passive + fair oracle: This passive baseline randomly selects points from the pool of examples $\mathcal{D}_{\text{tr}}^{\backslash y}$ and trains the model using the EFO oracle with the same $\alpha - \frac{1}{\sqrt{n}}$ constraint as FARE on its current samples.

Each of these methods with the exception of the passive baseline assumes access to a pretraining dataset. As we are interested in the low-data regime, when we do not have access to a pretraining dataset, we simulate the pretraining dataset by allocating, for each method, some percentage of the label budget to uniform sampling to collect a "pretrain" dataset, and then run the algorithm in standard fashion from there. For each method, we sweep over the size of the pretrain dataset and plot performance for the best one. For all other hyperparameters, we use the values recommended by the original work.

## 5.2 PERFORMANCE EVALUATION

We first consider the case when the fairness constraint is TPRP with $\alpha = 0.1$, and illustrate the accuracy and fairness vs. number of samples for our method and all baselines. For all methods and datasets, with the exception of PANDA, results are averaged over 100 trials—for PANDA results are averaged over only 50 trials, due to its large computational cost. Shaded regions denote one standard error. Note as well that the performance of PANDA starts at a later step since this method requires a large pretrain dataset to perform effectively, and in pretraining does not produce a classifier.

Our results are given in Figures 2 to 7, and we state the accuracy and fairness values obtained at the final step in Table 2. As these results illustrate, FARE consistently outperforms or matches the passive baseline, as well as

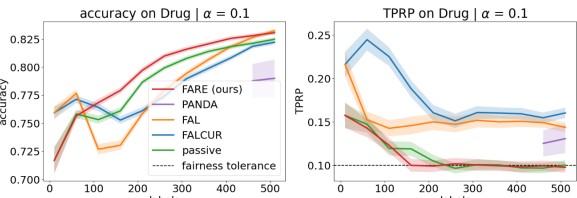

Figure 2: Performance on `Drug Consumption`

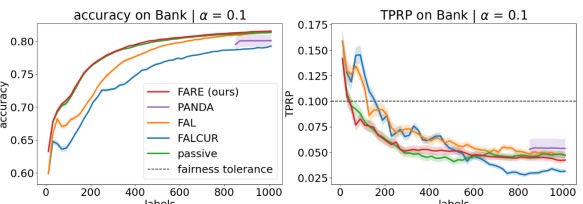

Figure 3: Performance on `Bank`

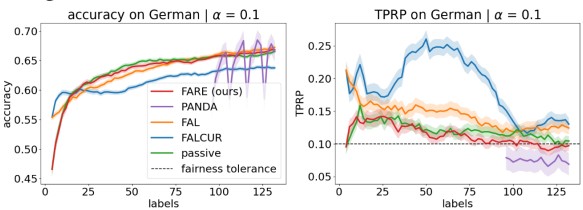

Figure 4: Performance on `German Credit`

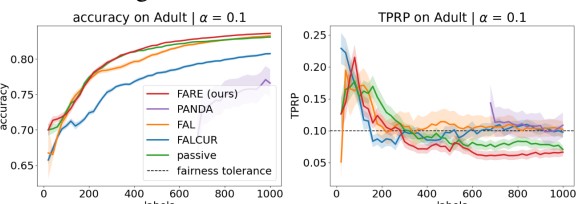

Figure 5: Performance on `Adult Income`

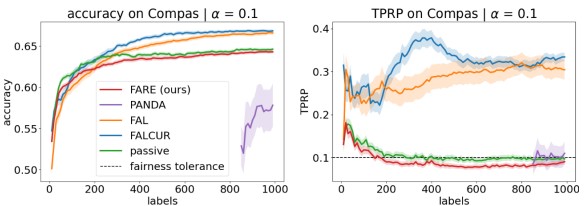

Figure 6: Performance on `Compas`

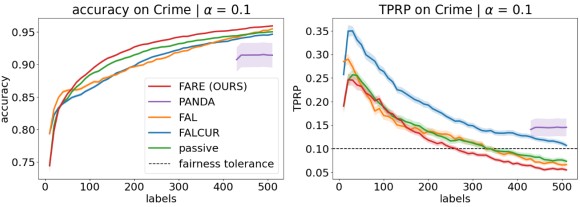

Figure 7: Performance on `Community and Crime`

|  | Accuracy (% labeled correctly) | | | | | Fairness (TPRP, goal fairness = 0.1) | | | | |
|---|---|---|---|---|---|---|---|---|---|---|
|  | FARE | PANDA | FAL | FALCUR | Passive | FARE | PANDA | FAL | FALCUR | Passive |
| Drug | **83.1** | 79.0 | 83.2 | 82.2 | 82.5 | 0.098 | 0.131 | 0.144 | 0.160 | 0.100 |
|  | **± 0.2** | ± 2.1 | ± 0.2 | ± 0.2 | ± 0.2 | ± 0.006 | ± 0.017 | ± 0.0065 | ± 0.006 | ± 0.005 |
| Bank | **81.5** | 80.1 | **81.3** | 79.2 | **81.3** | 0.042 | 0.054 | 0.047 | 0.032 | 0.047 |
|  | **± 0.1** | ± 0.4 | **± 0.1** | ± 0.1 | **± 0.1** | ± 0.003 | ± 0.009 | ± 0.003 | ± 0.002 | ± 0.001 |
| German | **66.8** | **66.4** | 67.2 | 63.7 | 66.6 | 0.097 | 0.069 | 0.124 | 0.130 | 0.104 |
|  | **± 0.3** | **± 1.4** | ± 0.4 | ± 0.4 | ± 0.3 | ± 0.007 | ± 0.016 | ± 0.010 | ± 0.010 | ± 0.007 |
| Adult | **83.6** | 76.6 | 83.2 | 80.8 | 83.1 | 0.065 | 0.109 | 0.102 | 0.097 | 0.068 |
|  | **± 0.0** | ± 2.0 | ± 0.1 | ± 0.2 | ± 0.0 | ± 0.007 | ± 0.019 | ± 0.013 | ± 0.008 | ± 0.006 |
| Compas | **64.3** | 57.8 | 66.6 | 66.8 | **64.6** | 0.088 | 0.110 | 0.304 | 0.334 | 0.099 |
|  | **± 0.1** | ± 1.3 | ± 0.2 | ± 0.2 | **± 0.2** | ± 0.006 | ± 0.026 | ± 0.023 | ± 0.009 | ± 0.007 |
| Crime | **95.9** | 91.4 | 95.5 | 94.7 | 95.0 | 0.055 | 0.145 | 0.066 | 0.107 | 0.074 |
|  | **± 0.1** | ± 1.7 | ± 0.1 | ± 0.1 | ± 0.1 | ± 0.004 | ± 0.019 | ± 0.004 | ± 0.005 | ± 0.005 |

Table 2: Final accuracy and TPRP values for each method and dataset. Blue indicates fairness threshold met, while red indicates threshold not met. Best accuracy among fair methods is indicated by **bold** font. Confidence intervals are standard errors based on 100 trials.

|  | Accuracy (% labeled correctly) | | | | | Fairness (TPRP, goal fairness = 0.1) | | | | |
|---|---|---|---|---|---|---|---|---|---|---|
|  | FARE | FARE w/o $\lambda_{\text{fair}}$ | FAL | FALCUR | Passive | FARE | FARE w/o $\lambda_{\text{fair}}$ | FAL | FALCUR | Passive |
| Synt. | **58.8** | 57.5 | 90.0 | 89.9 | 61.1 | 0.095 | 0.123 | 0.402 | 0.303 | 0.123 |
|  | **± 0.6** | ± 0.8 | ± 1.7 | ± 1.3 | ± 0.8 | ± 0.009 | ± 0.016 | ± 0.022 | ± 0.013 | ± 0.013 |

Table 3: Ablation on the role of group-dependent sampling, $\lambda_{\text{fair}}$, on the synthetically generated dataset. Note that PANDA does not converge on this dataset, so we have omitted it from the table. Confidence intervals are standard errors based on 100 trials.

all existing approaches to fair active classification. We highlight several key features of these results.

First, note that the only methods able to consistently produce classifiers which meet the fairness constraint of $\alpha = 0.1$ are FARE and the passive baselines. While all other methods frequently return classifiers that are unfair, both FARE and the passive baseline return classifiers that, by the final step, are fair on each dataset. We observe that, for very small number of labels, even FARE and the passive baselines produce classifiers which do not meet the fairness constraint—this is to be expected since, for a very small number of samples, it is difficult to estimate the fairness accurately enough to return a fair classifier. We emphasize that, though in some cases the accuracy of FARE is exceeded by baseline approaches, in most situations the baselines do not meet the fairness constraints. Since we are interested in *fair* classification, accuracy values can only be compared in the regime where each classifier is fair.

Second, we highlight the difference in the number of samples required to achieve a given accuracy for FARE as compared to the passive baseline. In particular, on the Drug, Adult, and Crime datasets, FARE requires between 1.4-2x less samples than passive to achieve the final accuracy achieved by passive, while ensuring the fairness constraint is still met. While this gain is not present on every dataset—for Bank and Compas the performance of FARE and the passive baseline are comparable—these results illustrate that active learning can yield substantial gains over passive approaches for fair classification, while simultaneously ensuring fairness constraints are met.

**Fairness Constraints Beyond TPRP.** The previously considered results illustrate the performance of each method when the fairness constraint is TPRP. To illustrate the generality of our approach, in Figure 8 we also consider the performance of each method when the fairness constraint is equalized odds. As with TPRP, we see that FARE produces a fair classifier while existing approaches fail to, and yields a marked improvement over the passive baseline in terms of accuracy.

**Model Selection Beyond Logistic Regression.** The aforementioned findings demonstrate how FARE performs when the model is a logistic regression classifier. To showcase the versatility of our method, in Figure 9, we compare FARE with passive when the model selection is a decision tree. Similar to logistic regression, we observe that FARE generates a fair classifier and yield a significant accuracy gain compared to the passive baseline.

## 5.3 ABLATION EXPERIMENTS

In this section, we illustrate the critical nature of two features of FARE. First, in Figure 10, we compare the performance of FARE with the fairness tolerance $\alpha - 1/\sqrt{n}$, with the $1/\sqrt{n}$ term correcting for the estimation error in the fairness constraint, to the performance with the fairness tolerance simply

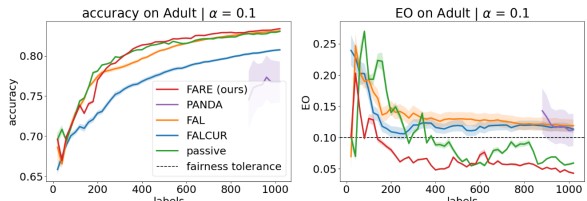

Figure 8: Performance on the Adult Income dataset for Equalized Odds

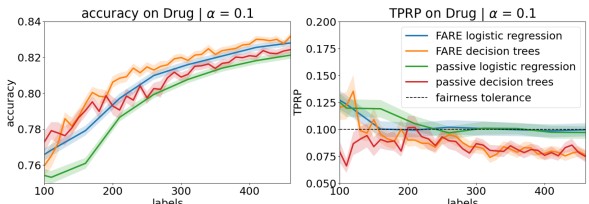

Figure 9: Performance on the Drug dataset with decision trees

set to $\alpha$. As shown, with the $1/\sqrt{n}$ correction, the classifier returned by FARE is unfair, while with the correction it is fair. We remark as well that, though the $1/\sqrt{n}$ correction is not precisely what is justified by Proposition 4.1, this value nonetheless consistently produces fair classifiers.

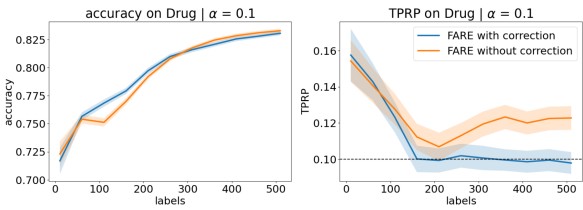

Figure 10: Ablation on fairness tolerance correction on Drug dataset

Lastly, in Table 3, we compare the performance of FARE with and without $\lambda_{\text{fair}}$, and additionally compare to the performance of the other baselines methods. We evaluate this on a synthetically generated dataset for which there is a large group imbalance—one group has significantly more examples in the dataset than the other. In this setting, if points are not explicitly sampled from the group with the smaller number of examples, virtually all samples will be taken from the larger group, which will cause the fairness estimates to be inaccurate, the resulting classifier unfair. This is illustrated in Table 3, where we see that without $\lambda_{\text{fair}}$, FARE produces an unfair classifier, similar to existing approaches. However, with $\lambda_{\text{fair}}$, FARE successfully achieves fairness. In conclusion, the inclusion of $\lambda_{\text{fair}}$ in FARE effectively ensures fairness constraints are met, especially when dealing with a significant group imbalance in the dataset.

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

## A  DATASETS DESCRIPTION

**Adult income dataset (Lichman, 2013)**: This dataset comprises $48,842$ examples with demographic information. The task is to predict whether an individual's income exceeds $50k\$$ annually. We chose the protected attribute to be binarized gender.

**Compas dataset (Lichman, 2013)**: This dataset, which was released by Angwin et al. (2022), encompasses $5,278$ data related to juvenile felonies. It includes details such as marital status, ethnicity, age, prior criminal history, and the severity of the current arrest charges. In our analysis, we identify binarized gender as a sensitive attribute. In line with established conventions (Anahideh et al., 2021; Corbett-Davies et al., 2017), we adopt a two-year violent recidivism record as the ground truth for assessing recidivism.

**Drug consumption dataset (Fehrman et al., 2017)**: This dataset consists of $1,885$ entries containing information about individuals, where each entry includes five demographic characteristics (such as Age, binarized Gender, or Education), seven measurements related to personality traits (such as Nscore indicating neuroticism and Ascore representing agreeableness), and 18 descriptors detailing the subject's most recent consumption of a specific substance (like Cannabis). We chose the task of predicting whether an individual consumed Cannabis in the last year and chose the protected attribute to be (binarized) Gender.

**German Credit dataset (Hofmann, 1994)**: The German Credit dataset classifies people as good or bad credit risks using the profile and history of $1,000$ clients. We set the binarized gender as the sensitive attribute.

**Community and Crime dataset (Redmond and Baveja, 2002)**: The Crime and Community dataset consists of $1,902$ instances of crimes with 128 attributes related to the crime and the corresponding community. It uses 'violent crimes' as the target variable and combines 'percentage of non-white' as the protected attribute. The target variable is binarized to categorize communities as high or low crime based on a threshold of $500$. The protected attribute is also binarized, separating communities with non-white residents below $20\%$.

**Bank dataset (Moro et al., 2014)**: The task is to predict whether the client has subscribed to a term deposit service based on $11,162$ data points with features such as marital status and age. We set the client having tertiary education as the sensitive attribute.

**Synthetic dataset**: We created the synthetic dataset in the following manner. It is depicted in Figure 11. The dataset consists of two dimensions, and data for group 0 is generated by randomly sampling $10,000$ data points from a Gaussian distribution with a mean of $(0,0)$, while group 1 comprises 100 data points sampled from $(10,10)$. For group 0 (and group 1), labels are assigned a value of 1 if the x-coordinate

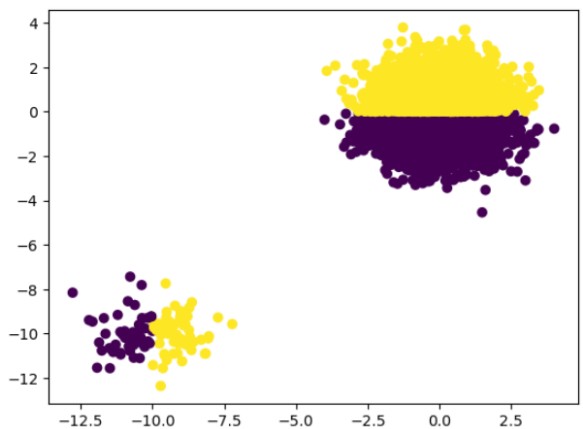

Figure 11: Synthetic dataset

(or y-coordinate) of the data point is greater than 0, and 0 otherwise. This ensures that each group is linearly separable, but their combination is not.

## B  PERFORMANCE OF BASELINE ALGORITHMS WITH DIFFERENT PRE-TRAINED DATASET SIZES

We report the results of the sweeps over the size of the pretrain dataset in Figures 12 to 23. Due to its large computational cost, we compared the performance of PANDA for two sizes of pretrain datasets.

## C  THEORETICAL RESULTS - PROOF OF PROPOSITION 4.1

### C.1  FULL THEOREM

We have the following result.

**Theorem C.1.** *Let the train set be* $\mathcal{D} = \{(x_1, a_1, y_1), \ldots, (x_n, a_n, y_n)\}$. *If* $\mathcal{D} \sim \nu$, *then it holds with probability* $1 - \delta$ *that:*

$$|L_\nu^{\text{EO}}(h) - \widehat{L}_\mathcal{D}^{\text{EO}}(h)| \le C_{0,0} + C_{0,1} + C_{1,0} + C_{1,1},$$
$$|L_\nu^{\text{TP}}(h) - \widehat{L}_\mathcal{D}^{\text{TP}}(h)| \le C_{0,1} + C_{1,1},$$
$$|L_\nu^{\text{FP}}(h) - \widehat{L}_\mathcal{D}^{\text{FP}}(h)| \le C_{0,0} + C_{1,0},$$

*with confidence terms*

$$C_{j,k} = \left( \widehat{p}_{j,k} + \sqrt{2\widehat{\mathcal{V}}_{j,k}^{(1)} \frac{\log(2/\delta)}{n}} + \frac{\log(2/\delta)}{n} \right) \times$$
$$\times \frac{\sqrt{2\widehat{\mathcal{V}}_{j,k}^{(2)} \frac{\log(2/\delta)}{n}} + \frac{\log(2/\delta)}{n}}{\left( \frac{1}{n} \sum_{i=1}^n \mathbb{1}\{y_i = k, a_i = j\} \right)^2}$$
$$+ \frac{\sqrt{2\widehat{\mathcal{V}}_{j,k}^{(1)} \frac{\log(2/\delta)}{n}} + \frac{\log(2/\delta)}{n}}{\frac{1}{n} \sum_{i=1}^n \mathbb{1}\{y_i = k, a_i = j\}}$$

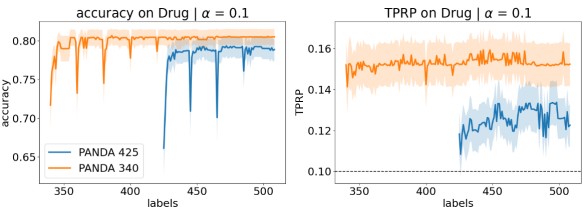

Figure 12: Performance on `Drug Consumption`

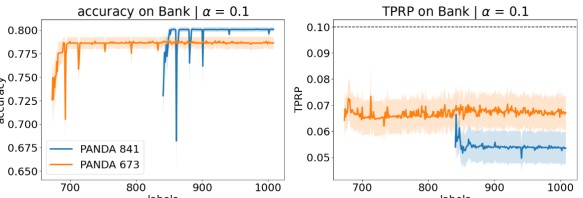

Figure 13: Performance on `Bank`

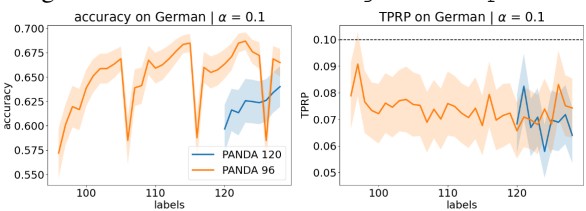

Figure 14: Performance on `German Credit`

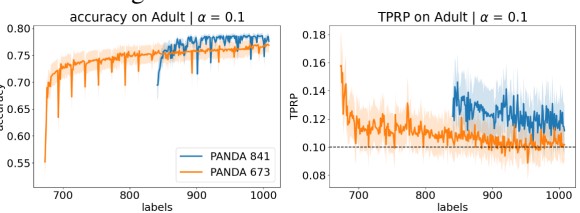

Figure 15: Performance on `Adult Income`

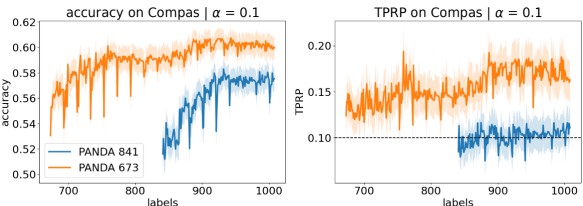

Figure 16: Performance on `Compas`

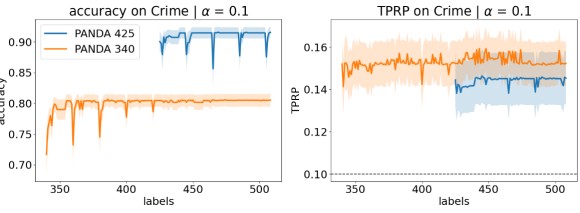

Figure 17: Performance on `Community and Crime`

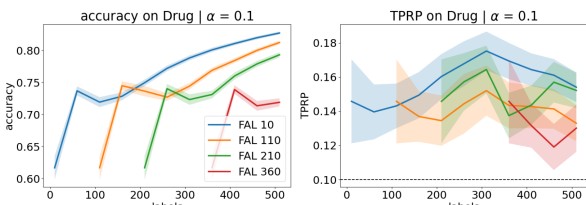

Figure 18: Performance on `Drug Consumption`

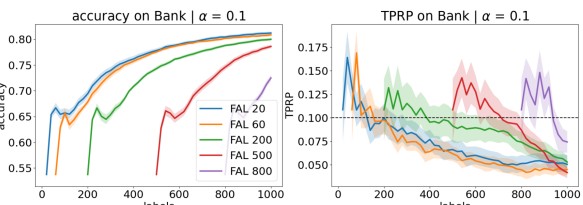

Figure 19: Performance on `Bank`

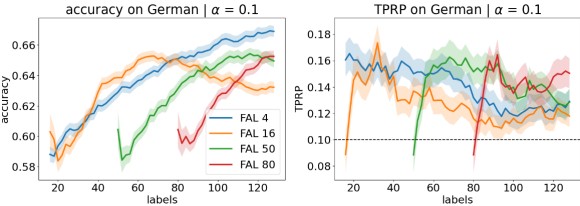

Figure 20: Performance on `German Credit`

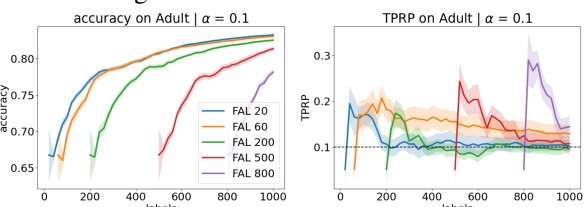

Figure 21: Performance on `Adult Income`

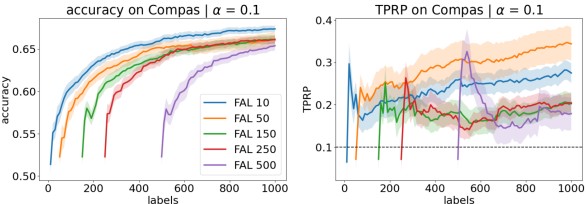

Figure 22: Performance on `Compas`

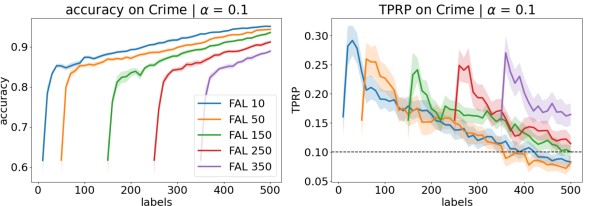

Figure 23: Performance on `Community and Crime`

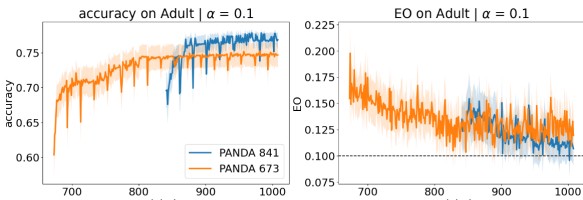

Figure 24: Performance on `Adult Income` for Equalized Odds

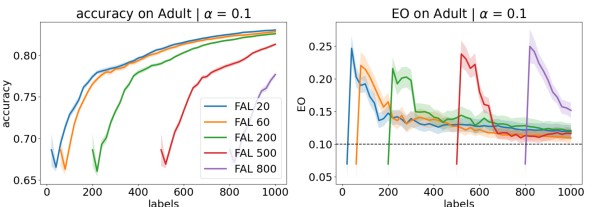

Figure 25: Performance on `Adult Income` for Equalized Odds

*for label $k \in \{0,1\}$ and protected attribute $j \in \{0,1\}$, where $\widehat{p}_{j,k} = \frac{1}{n}\sum_{i=1}^n \mathbb{1}\{h(x_i) = 1, y_i = k, a_i = j\}$ and the empirical variances defined as*

$$
\widehat{\mathcal{V}}_{j,k}^{(1)} = \frac{1}{n(n-1)} \sum_{1 \leq \ell < \ell' \leq n} (\mathbb{1}\{h(x_\ell) = 1, y_\ell = k, a_\ell = j\}
$$
$$
- \mathbb{1}\{h(x_{\ell'}) = 1, y_{\ell'} = k, a_{\ell'} = j\})^2,
$$
$$
\widehat{\mathcal{V}}_{j,k}^{(2)} = \frac{1}{n(n-1)} \sum_{1 \leq \ell < \ell' \leq n} (\mathbb{1}\{y_\ell = k, a_\ell = j\}
$$
$$
- \mathbb{1}\{y_{\ell'} = k, a_{\ell'} = j\})^2.
$$

This theorem provides a confidence bound on the concentration rate of the empirical fairness violation.

*Proof.* Let us start by proving the statement for TPRP. Recall

$$
L_\nu^{\mathrm{TP}}(h) = \left| \frac{P_{(x,a,y)\sim\nu}(h(x) = 1, a = 0, y = 1)}{P_{(x,a,y)\sim\nu}(a = 0, y = 1)} \right.
$$
$$
\left. - \frac{P_{(x,a,y)\sim\nu}(h(x) = 1, a = 1, y = 1)}{P_{(x,a,y)\sim\nu}(a = 1, y = 1)} \right|
$$
$$
\widehat{L}_\mathcal{D}^{\mathrm{TP}}(h) = \left| \sum_{i=1}^n \frac{\mathbb{1}\{h(x_i) = 1, y_i = 1, a_i = 1\}}{\sum_{i=1}^n \mathbb{1}\{y_i = 1, a_i = 1\}} \right.
$$
$$
\left. - \sum_{i=1}^n \frac{\mathbb{1}\{h(x_i) = 1, y_i = 1, a_i = 0\}}{\sum_{i=1}^n \mathbb{1}\{y_i = 1, a_i = 0\}} \right|.
$$

and write these for short

$$
L_\nu^{\mathrm{TP}}(h) = |\mathrm{num}_0/\mathrm{den}_0 - \mathrm{num}_1/\mathrm{den}_1|,
$$
$$
\widehat{L}_\mathcal{D}^{\mathrm{TP}}(h) = |\widehat{\mathrm{num}}_0/\widehat{\mathrm{den}}_0 - \widehat{\mathrm{num}}_1/\widehat{\mathrm{den}}_1|,
$$

with for protected attribute $j \in \{0,1\}$,

$$
\mathrm{num}_j = P_{(x,a,y)\sim\nu}(h(x) = 1, a = j, y = 1)
$$
$$
\widehat{\mathrm{num}}_j = \frac{1}{n}\sum_{i=1}^n \mathbb{1}\{h(x_i) = 1, y_i = 1, a_i = j\}
$$
$$
\mathrm{den}_j = P_{(x,a,y)\sim\nu}(a = j, y = 1)
$$
$$
\widehat{\mathrm{den}}_j = \frac{1}{n}\sum_{i=1}^n \mathbb{1}\{y_i = 1, a_i = j\}.
$$

Applying Bernstein's concentration bound it holds that for $j \in \{0,1\}$ with probability at least $1 - \delta$

$$
|\widehat{\mathrm{num}}_j - \mathrm{num}_j| = \left| \frac{1}{n}\sum_{i=1}^n \mathbb{1}\{h(x_i) = 1, y_i = 1, a_i = j\} \right.
$$
$$
\left. - \mathbb{P}_{(x,a,y)\sim\nu}(h(x) = 1, y = 1, a = j) \right|
$$
$$
\leq \sqrt{2\widehat{\mathcal{V}}_{j,1}^{(1)}\frac{\log(2/\delta)}{n}} + \frac{\log(2/\delta)}{n} =: \alpha_j^{(\mathrm{num})},
$$

where we defined

$$
\widehat{\mathcal{V}}_{j,k}^{(1)} = \frac{1}{n(n-1)} \sum_{1 \leq \ell < \ell' \leq n} (\mathbb{1}\{h(x_\ell) = 1, y_\ell = k, a_\ell = j\}
$$
$$
- \mathbb{1}\{h(x_{\ell'}) = 1, y_{\ell'} = k, a_{\ell'} = j\})^2.
$$

Also applying Bernstein's concentration bound it holds that for $j \in \{0,1\}$ with probability at least $1 - \delta$

$$
|\widehat{\mathrm{den}}_j - \mathrm{den}_j| = \left| \frac{1}{n}\sum_{i=1}^n \mathbb{1}\{y_i = 1, a_i = j\} - \right.
$$
$$
\left. \mathbb{P}_{(x,a,y)\sim\nu}(y = 1, a = j) \right|
$$
$$
\leq \sqrt{2\widehat{\mathcal{V}}_{j,1}^{(2)}\frac{\log(2/\delta)}{n}} + \frac{\log(2/\delta)}{n} =: \alpha_j^{(\mathrm{den})},
$$

where we defined

$$
\widehat{\mathcal{V}}_{j,k}^{(2)} = \frac{1}{n(n-1)} \sum_{1 \leq \ell < \ell' \leq n} (\mathbb{1}\{y_\ell = k, a_\ell = j\}
$$
$$
- \mathbb{1}\{y_{\ell'} = k, a_{\ell'} = j\})^2.
$$

Then, as soon as for both $j = 1$ and $j = 2$, $\alpha_j^{(\mathrm{den})} \leq \widehat{\mathrm{den}}_j/2$, holds the inequality

$$
\left| \frac{1}{\widehat{\mathrm{den}}_j} - \frac{1}{\mathrm{den}_j} \right| \leq \frac{\alpha_j^{(\mathrm{den})}}{\widehat{\mathrm{den}}_j^2},
$$

so that for $j \in \{0, 1\}$, we have

$$
\begin{aligned}
\left| \frac{\widehat{\text{num}}_j}{\widehat{\text{den}}_j} - \frac{\text{num}_j}{\text{den}_j} \right| &= \left| \frac{\widehat{\text{num}}_j}{\widehat{\text{den}}_j} - \frac{\text{num}_j}{\widehat{\text{den}}_j} - \frac{\text{num}_j}{\widehat{\text{den}}_j} - \frac{\text{num}_j}{\text{den}_j} \right| \\
&\leq \left| \frac{\widehat{\text{num}}_j}{\widehat{\text{den}}_j} - \frac{\text{num}_j}{\widehat{\text{den}}_j} \right| + \left| \frac{\text{num}_j}{\widehat{\text{den}}_j} - \frac{\text{num}_j}{\text{den}_j} \right| \\
&\leq \frac{\alpha_j^{(\text{num})}}{\widehat{\text{den}}_j} + \frac{\text{num}_j \alpha_j^{(\text{den})}}{\widehat{\text{den}}_j^2} \\
&\leq \frac{\alpha_j^{(\text{num})}}{\widehat{\text{den}}_j} + \frac{(\alpha_j^{(\text{num})} + \widehat{\text{num}}_j) \alpha_j^{(\text{den})}}{\widehat{\text{den}}_j^2}.
\end{aligned}
$$

Note that $C_{j,1}$ is exactly the last upper bound above,

$$
\begin{aligned}
C_{j,1} =& \left( \widehat{p}_{j,1} + \sqrt{2 \widehat{\mathcal{V}}_{j,1}^{(1)} \frac{\log(2/\delta)}{n}} + \frac{\log(2/\delta)}{n} \right) \times \\
& \times \frac{\sqrt{2 \widehat{\mathcal{V}}_{j,1}^{(2)} \frac{\log(2/\delta)}{n}} + \frac{\log(2/\delta)}{n}}{\left( \frac{1}{n} \sum_{i=1}^n \mathbb{1}\{y_i = 1, a_i = j\} \right)^2} \\
& + \frac{\sqrt{2 \widehat{\mathcal{V}}_{j,1}^{(1)} \frac{\log(2/\delta)}{n}} + \frac{\log(2/\delta)}{n}}{\frac{1}{n} \sum_{i=1}^n \mathbb{1}\{y_i = 1, a_i = j\}}
\end{aligned}
$$

where $\widehat{p}_{j,1} = \frac{1}{n} \sum_{i=1}^n \mathbb{1}\{h(x_i) = 1, y_i = 1, a_i = j\}$. Putting it together

$$
\begin{aligned}
|L_\nu^{\text{TP}}(h) - \widehat{L}_{\mathcal{D}}^{\text{TP}}(h)| =& ||\text{num}_0/\text{den}_0 - \text{num}_1/\text{den}_1| \\
& - |\widehat{\text{num}}_0/\widehat{\text{den}}_0 - \widehat{\text{num}}_1/\widehat{\text{den}}_1||, \\
\leq& |\text{num}_0/\text{den}_0 - \text{num}_1/\text{den}_1 \\
& - \widehat{\text{num}}_0/\widehat{\text{den}}_0 + \widehat{\text{num}}_1/\widehat{\text{den}}_1|, \\
\leq& |\text{num}_0/\text{den}_0 - \widehat{\text{num}}_0/\widehat{\text{den}}_0| \\
& + |\widehat{\text{num}}_1/\widehat{\text{den}}_1 - \text{num}_1/\text{den}_1|, \\
\leq& C_{0,1} + C_{1,1}.
\end{aligned}
$$

which is the conclusion for TPRP.

As $\widehat{L}_{\mathcal{D}}^{\text{FP}}(h)$ was defined as the empirical estimate of the FPRP violation by conditioning on $\mathbb{1}\{y_i = 0\}$ (instead of $\mathbb{1}\{y_i = 1\}$ for TPRP), the proof for the concentration bound on FPRP is analogous to the one of TPRP, with the exception of the conditioning on $\mathbb{1}\{y_i = 0\}$ instead of $\mathbb{1}\{y_i = 1\}$ for TPRP.

We defined the empirical estimate of the EO violation as the maximum of empirical estimate of the TPRP violation and the empirical estimate of the FPRP violation, $\widehat{L}_{\mathcal{D}}^{\text{EO}}(h) = \max\{\widehat{L}_{\mathcal{D}}^{\text{TP}}(h), \widehat{L}_{\mathcal{D}}^{\text{FP}}(h)\}$, so holds

$$
\widehat{L}_{\mathcal{D}}^{\text{EO}}(h) \leq \widehat{L}_{\mathcal{D}}^{\text{TP}}(h) + \widehat{L}_{\mathcal{D}}^{\text{FP}}(h),
$$

which immediately leads to the conclusion of Theorem C.1.
□

## C.2 PROOF OF PROPOSITION 4.1

We first state the full result that leads to the statement of Proposition 4.1.

**Proposition C.2.** *Let the train set be* $\mathcal{D} = \{(x_1, a_1, y_1), \ldots, (x_n, a_n, y_n)\}$. *If* $\mathcal{D} \sim \nu$, *then it holds with probability* $1 - \delta$ *that:*

$$
|L_\nu^{\text{TP}}(h) - \widehat{L}_{\mathcal{D}}^{\text{TP}}(h)| \leq
$$
$$
2 \max_{j \in \{0,1\}} \left\{ 2 \left( \frac{\sqrt{2 \frac{\log(2/\delta)}{n}} + \frac{\log(2/\delta)}{n}}{\frac{1}{n} \sum_{i=1}^n \mathbb{1}\{y_i = 1, a_i = j\}} \right) \right.
$$
$$
\left. + \left( \frac{\sqrt{2 \frac{\log(2/\delta)}{n}} + 2 \frac{\log(2/\delta)}{n}}{\frac{1}{n} \sum_{i=1}^n \mathbb{1}\{y_i = 1, a_i = j\}} \right)^2 \right\},
$$
$$
|L_\nu^{\text{EO}}(h) - \widehat{L}_{\mathcal{D}}^{\text{EO}}(h)| \leq
$$
$$
4 \max_{0 \leq j,k \leq 1} \left\{ 2 \left( \frac{\sqrt{2 \frac{\log(2/\delta)}{n}} + \frac{\log(2/\delta)}{n}}{\frac{1}{n} \sum_{i=1}^n \mathbb{1}\{y_i = k, a_i = j\}} \right) \right.
$$
$$
\left. + \left( \frac{\sqrt{2 \frac{\log(2/\delta)}{n}} + 2 \frac{\log(2/\delta)}{n}}{\frac{1}{n} \sum_{i=1}^n \mathbb{1}\{y_i = k, a_i = j\}} \right)^2 \right\}.
$$

*Proof of Proposition 4.1 and C.2.* We use Theorem C.1 and for label $k \in \{0, 1\}$ and protected attribute $j \in \{0, 1\}$ we bound $C_{j,k}$.

We first have that the empirical variances are such that $\widehat{\mathcal{V}}_{j,k}^{(1)} \leq 1$ and $\widehat{\mathcal{V}}_{j,k}^{(2)} \leq 1$. Also,

$$
\begin{aligned}
\widehat{p}_{j,k} &= \frac{1}{n} \sum_{i=1}^n \mathbb{1}\{h(x_i) = 1, y_i = k, a_i = j\} \\
&\leq \frac{1}{n} \sum_{i=1}^n \mathbb{1}\{y_i = k, a_i = j\}.
\end{aligned}
$$

Thus, we can bound

$$
\begin{aligned}
C_{j,k} =& \left( \widehat{p}_{j,k} + \sqrt{2 \widehat{\mathcal{V}}_{j,k}^{(1)} \frac{\log(2/\delta)}{n}} + \frac{\log(2/\delta)}{n} \right) \times \\
& \times \frac{\sqrt{2 \widehat{\mathcal{V}}_{j,k}^{(2)} \frac{\log(2/\delta)}{n}} + \frac{\log(2/\delta)}{n}}{\left( \frac{1}{n} \sum_{i=1}^n \mathbb{1}\{y_i = k, a_i = j\} \right)^2} \\
& + \frac{\sqrt{2 \widehat{\mathcal{V}}_{j,k}^{(1)} \frac{\log(2/\delta)}{n}} + \frac{\log(2/\delta)}{n}}{\frac{1}{n} \sum_{i=1}^n \mathbb{1}\{y_i = k, a_i = j\}} \\
\leq& 2 \left( \frac{\sqrt{2 \frac{\log(2/\delta)}{n}} + \frac{\log(2/\delta)}{n}}{\frac{1}{n} \sum_{i=1}^n \mathbb{1}\{y_i = k, a_i = j\}} \right) \\
& + \left( \frac{\sqrt{2 \frac{\log(2/\delta)}{n}} 2 \frac{\log(2/\delta)}{n}}{\frac{1}{n} \sum_{i=1}^n \mathbb{1}\{y_i = k, a_i = j\}} \right)^2.
\end{aligned}
$$

With that result, we conclude for TPRP that

$$|L_\nu^{\mathrm{TP}}(h) - \widehat{L}_D^{\mathrm{TP}}(h)|$$
$$\leq C_{0,1} + C_{1,1}$$
$$\leq 2 \max_{j \in \{0,1\}} C_{j,1}$$
$$\leq 2 \max_{j \in \{0,1\}} \left\{ 2 \left( \frac{\sqrt{2\frac{\log(2/\delta)}{n}} + \frac{\log(2/\delta)}{n}}{\frac{1}{n}\sum_{i=1}^n \mathbb{1}\{y_i = 1, a_i = j\}} \right) \right.$$
$$\left. + \left( \frac{\sqrt{2\frac{\log(2/\delta)}{n}} + 2\frac{\log(2/\delta)}{n}}{\frac{1}{n}\sum_{i=1}^n \mathbb{1}\{y_i = 1, a_i = j\}} \right)^2 \right\}$$
$$= 4 \max_{j \in \{0,1\}} \frac{\sqrt{2\frac{\log(2/\delta)}{n}}}{\frac{1}{n}\sum_{i=1}^n \mathbb{1}\{y_i = 1, a_i = j\}} + \mathcal{O}\left( \frac{1}{n} \right).$$

Analogous bounds conclude for EO:

$$|L_\nu^{\mathrm{EO}}(h) - \widehat{L}_D^{\mathrm{EO}}(h)|$$
$$\leq C_{0,0} + C_{1,0} + C_{0,1} + C_{1,1}$$
$$\leq 4 \max_{j \in \{0,1\}} C_{j,k}$$
$$\leq 4 \max_{0 \leq j,k \leq 1} \left\{ 2 \left( \frac{\sqrt{2\frac{\log(2/\delta)}{n}}\frac{\log(2/\delta)}{n}}{\frac{1}{n}\sum_{i=1}^n \mathbb{1}\{y_i = k, a_i = j\}} \right) \right.$$
$$\left. + \left( \frac{\sqrt{2\frac{\log(2/\delta)}{n}} + 2\frac{\log(2/\delta)}{n}}{\frac{1}{n}\sum_{i=1}^n \mathbb{1}\{y_i = k, a_i = j\}} \right)^2 \right\}$$
$$= 8 \max_{0 \leq j,k \leq 1} \frac{\sqrt{2\frac{\log(2/\delta)}{n}}}{\frac{1}{n}\sum_{i=1}^n \mathbb{1}\{y_i = k, a_i = j\}} + \mathcal{O}\left( \frac{1}{n} \right).$$

$\square$