# OpenReview forum: "Fair Active Learning in Low-Data Regimes"
_auai.org/UAI/2024/Conference — UAI 2024 poster_

### Official Review · Reviewer_6Sz8 · 2024-03-09

**Q2-1 Originality-Novelty:** 2
**Q2-2 Correctness-Technical Quality:** 3
**Q2-5 Clarity Of Writing:** 2

**Q1 Summary And Contributions:**

This paper presents a new fair active learning algorithm for group fairness and empirically shows it outperforms several baseline fair active learning baseline methods on real-world data sets.

**Q2-3 Extent To Which Claims Are Supported By Evidence:**

3: Good: the main claims are supported by convincing evidence (in the form of adequate experimental evaluation, proofs, (pseudo-)code, references, assumptions).

**Q2-4 Reproducibility:**

3: Good: key resources (e.g. proofs, code, data) are available and key details (e.g. proofs, experimental setup) are sufficiently well-described for competent researchers to confidently reproduce the main results.

**Q3 Main Strengths:**

S1. I think addressing fairness in the active learning setting remains an important problem.

**Q4 Main Weakness:**

W1. Novelty of the proposed method is limited, because it is basically an existing accuracy-oriented AL algorithm which incorporates fairness by labeling equal number of instances from each demographic group.

W2. Problem setting is not clearly stated. For example, does the paper consider active labeling of given unlabeled data or active sampling from a given data distribution? Also, does the AL algorithm aim to efficiently improve accuracy (while maintaining fairness) or efficiently improve fairness (while maintaining accuracy)? Please clarify.

W3. Description of the proposed fair AL algorithm is not clear, and there are only two short paragraphs describing it (at the end of Section 4). Please elaborate e.g., on how the proposed algorithm improves algorithm (and provide proper/deeper justification if possible).

W4. The proposed AL algorithm resembles a notable line of technique called "disagreement-based AL", but discussion and comparison with this related literature is completely missing. Please add them.

**Q5 Detailed Comments To The Authors:**

The paper has some interesting idea, but novelty seems a bit limited and important related work on AL is missing in discussion. Also, clarity of the problem setting and proposed algorithm should be improved, and deeper investigation/explanation on how the proposed algorithm improves fairness is missing but suggested to be performed (shouldn't be too hard based on my educated guess).

Overall, the work is not ready for publishing at UAI.

**Q9 Complying With Reviewing Instructions:**

Yes

---

### Official Review · Reviewer_oBCp · 2024-03-21

**Q2-1 Originality-Novelty:** 2
**Q2-2 Correctness-Technical Quality:** 2
**Q2-5 Clarity Of Writing:** 3

**Q1 Summary And Contributions:**

This paper consider the active learning setting under fairness constraint.
More precisely, the authors focus on binary classification with active sampling.
In this context, the objective is to build classifier that satisfies fairness constraint. In the paper, two notion of fairness are investigated (equal opportunity
and equalized odds).
The proposed procedure relies the combination of two methods (one to enforce fairness, the other for the active sampling). In particular, the authors consider an in-processing methods to enforce fairness as in [1], or [2].
Then the active sampling is based on the works of and involves the random exploration principle.
The methods is evaluated on both synthetic and real data and is compared to a benchmark. Notably, the procedure exhibits good performance.

**Q2-3 Extent To Which Claims Are Supported By Evidence:**

2: Fair: the main claims are somewhat supported by evidence (but the experimental evaluation may be weak, or does not match entirely with the claims, important baselines may be missing, proofs contain important ideas but lack rigor, algorithmic details are only discussed superficially, references are imprecise, assumptions are not sufficiently motivated or explicated, etc.).

**Q2-4 Reproducibility:**

2: Fair: key resources (e.g. proofs, code, data) are unavailable but key details (e.g. proof sketches, experimental setup) are sufficiently well-described for an expert to confidently reproduce the main results.

**Q3 Main Strengths:**

The paper is clearly written and the proposed procedure is well presented.
In particular, it nicely convey how to combine methods dedicated respectively to enforce fairness and for the active sampling.
One of the main strength of the paper is that the proposed algorithm seems to have good numerical performance. However, the procedure is only presented for the logistic regression.

**Q4 Main Weakness:**

A possible limitation of the method is that the author only consider the logistic regression in their numerical study. I think that other algorithm should be investigated to assess the quality of the procedure.

Another limitation is that the authors does not provide finite sample bound for the procedure. For instance, such bound could be compared with the passive rate and the existing literature.

**Q5 Detailed Comments To The Authors:**

The paper is clearly written and the proposed procedure is well presented.
In particular, it nicely convey how to combine methods dedicated respectively to enforce fairness and for the active sampling.
One of the main strength of the paper is that the proposed algorithm seems to have good numerical performance. However, the procedure is only presented for the logistic regression.

1) a possible limitation of the method is that the author only consider the logistic regression in their numerical study. I think that other algorithm should be investigated to assess the quality of the procedure.

2) Another limitation is that the authors does not provide finite sample bound for the procedure. For instance, such bound could be compared with the passive rate and the existing literature.

3) I wonder if the procedure can be easily extend to multi-valued sensitive attribute.
It could improve the impact of the paper.


4) Finally, a minor point, in general considering demographic parity constraint is more easy. Perhaps, the authors can add a remark to the extension of their procedure to demographic parity constraint.


[1] Donini, M., Oneto, L., Ben-David, S., Shawe-Taylor,J. S., and Pontil, M. (2018). Empirical risk minimization under fairness constraints.

[2] Agarwal, A., Beygelzimer, A., Dudík, M., Langford, J., and Wallach, H. . A reductions approach to fair classification.

**Q9 Complying With Reviewing Instructions:**

Yes

---

### Official Review · Reviewer_jPc1 · 2024-03-23

**Q2-1 Originality-Novelty:** 2
**Q2-2 Correctness-Technical Quality:** 3
**Q2-5 Clarity Of Writing:** 4

**Q1 Summary And Contributions:**

This paper provides active learning algorithms that achieves group fairness. Detailed theoretical and empirical analysis are presented.

**Q2-3 Extent To Which Claims Are Supported By Evidence:**

3: Good: the main claims are supported by convincing evidence (in the form of adequate experimental evaluation, proofs, (pseudo-)code, references, assumptions).

**Q2-4 Reproducibility:**

3: Good: key resources (e.g. proofs, code, data) are available and key details (e.g. proofs, experimental setup) are sufficiently well-described for competent researchers to confidently reproduce the main results.

**Q3 Main Strengths:**

The presentation of this paper is neat and organized. It clearly explains how their work differentiates from others in the field and the technical details seem solid. The tradeoff between accuracy metric and fairness metric are properly shown.

**Q4 Main Weakness:**

Active learning naturally implies very little data needs to be used for training, so emphasizing the same point twice in the title seems a bit excessive but this might just be my personal preference.

**Q5 Detailed Comments To The Authors:**

It may be worth it to spend some time discussing why active vs passive doesn't make a big difference for fairness metric in Figure 2 and Figure 4 .

**Q9 Complying With Reviewing Instructions:**

Yes

---

### Official Review · Reviewer_YVDo · 2024-03-23

**Q2-1 Originality-Novelty:** 4
**Q2-2 Correctness-Technical Quality:** 3
**Q2-5 Clarity Of Writing:** 4

**Q1 Summary And Contributions:**

The authors propose a novel active learning framework which uses a sampling procedure along with a clasification procedure. The framework is effective in the domains of data scaricity to maximise accuracy. The framework also satisfies fairness constraints with high prpobability. The paper also consolidates sufficient experiments to justify their claims.

**Q2-3 Extent To Which Claims Are Supported By Evidence:**

4: Excellent: all claims are supported by very convincing evidence (in the form of comprehensive experimental evaluation, rigorous mathematical proofs, detailed (pseudo-)code, precise references, well-motivated and realistic assumptions) and the authors deliver what they promise.

**Q2-4 Reproducibility:**

3: Good: key resources (e.g. proofs, code, data) are available and key details (e.g. proofs, experimental setup) are sufficiently well-described for competent researchers to confidently reproduce the main results.

**Q3 Main Strengths:**

1. The paper is well written and easy to follow.
2. The paper discusses a cruicial issue of fairness in low data regimes
3. The study throughly covers all the related works and sufficient introduction is included.

**Q4 Main Weakness:**

1. I am interested to know if the authors work with image of other non tabular dataset.
2. Did the authors explore any other technique apart from randomized exploration.

**Q5 Detailed Comments To The Authors:**

Please see above.

**Q9 Complying With Reviewing Instructions:**

Yes

---

### Official Review · Reviewer_sLbk · 2024-03-25

**Q2-1 Originality-Novelty:** 2
**Q2-2 Correctness-Technical Quality:** 2
**Q2-5 Clarity Of Writing:** 3

**Q1 Summary And Contributions:**

The paper considers fairness in the context of active learning. The paper considers binary classification with one binary-valued protected feature, and focuses on Equal Opportunity and Equalized Odds notions of fairness. The paper proposes to consider active learning for the efficient utilization of the data (or potential collection). In particular, a group-dependent posterior sampling approach is presented, with emphases on the bound over empirical violation of fairness.

**Q2-3 Extent To Which Claims Are Supported By Evidence:**

2: Fair: the main claims are somewhat supported by evidence (but the experimental evaluation may be weak, or does not match entirely with the claims, important baselines may be missing, proofs contain important ideas but lack rigor, algorithmic details are only discussed superficially, references are imprecise, assumptions are not sufficiently motivated or explicated, etc.).

**Q2-4 Reproducibility:**

3: Good: key resources (e.g. proofs, code, data) are available and key details (e.g. proofs, experimental setup) are sufficiently well-described for competent researchers to confidently reproduce the main results.

**Q3 Main Strengths:**

The strength of the paper comes from the effort to address the relation between accurate and fair classifier and sample size, from the active learning perspective. Overall, the paper is easy to follow. The problem setting, the proposed approach (algorithm and theoretical characterization), and empirical results are presented relatively clear.

**Q4 Main Weakness:**

The weakness of the paper comes from following points/potential concerns:

- the theoretical analysis is a direct result of Bernstein's concentration bound, what is the relation between the theoretical analysis to the active learning objective?

- what is the takeaway message of the proposed approach, how to generalize/extend in order to apply to cases beyond binary classification with a single binary protected feature, with different fairness notions?

- the potential tension between unbiased estimation of fairness violation and active learning

**Q5 Detailed Comments To The Authors:**

**w.r.t. the theoretical result Proposition 4.1**

The theoretical result (Proposition 4.1) presents the convergence behavior of the gap of true fairness violation and its empirical estimation. As presented in Appendix C, Proposition 4.1 is a result of directly applying Bernstein's concentration bound multiple times (number of classes squared $\times$ cardinality of protected feature, both of which are binary in the paper). What information does this bound provide w.r.t. "ensuring the desired fairness tolerance" compared to previous works involving active learning? Further clarifications are needed to back the claim that the proposed approach address this limitation of previous works.

**w.r.t. potential generalization and takeaway msg of the work**

The paper focuses on binary classification with one binary-valued protected feature. The algorithm and theoretical result are tailored to this setting. When going beyond binary cases, one can image things can get complicated. How to generalize and extend the current framework? Furthermore, the paper focuses on Equal Opportunity and Equalized Odds notions of fairness and claims that "method extends to other notions of fairness as well" (Section 3.1). This does not seem to be the case, e.g., for Predictive Rate Parities (another widely used group-level fairness notion), where $\hat{y}$ (instead of $y$) is conditioned on.

**the potential tension between unbiased estimation of fairness violation and active learning**

The paper proposes to consider equal number of samples from each group for unbiased empirical fairness violation (Section 4.3). This does not necessarily align with finding the most informative data points from the active learning perspective. Can authors comment on this potential tension, and its implication on the overall approach?

**Q9 Complying With Reviewing Instructions:**

Yes

---

### Meta-Review · Area_Chair_pTGK · 2024-04-18

The paper proposes a method for active learning under fairness constraints. A finite-sample theoretical analysis of learning under fairness constraints motivates some of the design choices. The proposed algorithm alternates between acquiring using two different sampling distributions, motivated by the accuracy and fairness constraints, respectively, while correcting to some degree for the sampling bias inherent in active learning. The experimental analysis is solid and shows the proposed method excels particularly in the low-data regime, which is the main claimed purpose of the work.
Reviewers were somewhat worried about novelty and about the lack of theoretical analysis of the overall algorithm. However, the paper addresses an important problem which hasn't been widely studied; proposes a new and well-motivated algorithm; has good experimental analysis results; and is well-written. While an overall theoretical analysis would be great, the current extent of the paper's content is sufficient.